# Bibliometric Analysis and a Call for Increased Rigor in Citing Scientific Literature: Folic Acid Fortification and Neural Tube Defect Risk as an Example

**DOI:** 10.3390/nu16152503

**Published:** 2024-08-01

**Authors:** Brynne Boeck, Cara J. Westmark

**Affiliations:** 1Department of Neurology, University of Wisconsin, Madison, WI 53706, USA; bboeck@wisc.edu; 2Molecular Environmental Toxicology Center, University of Wisconsin, Madison, WI 53706, USA

**Keywords:** bibliometrics, folate, folic acid, food fortification, neural tube defects, rigor and reproducibility

## Abstract

The health benefits of vitamin B9 (folate) are well documented, particularly in regard to neural tube defects during pregnancy; however, much remains to be learned regarding the health effects and risks of consuming folic acid supplements and foods fortified with folic acid. In 2020, our laboratory conducted a population-based analysis of the Food Fortification Initiative (FFI) dataset to determine the strength of the evidence regarding the prevalence of neural tube defects (NTD) at the national level in response to mandatory fortification of cereal grains with folic acid. We found a very weak correlation between the prevalence of NTDs and the level of folic acid fortification irrespective of the cereal grain fortified (wheat, maize, or rice). We found a strong linear relationship between reduced NTDs and higher socioeconomic status (SES). Our paper incited a debate on the proper statistics to employ for population-level data. Subsequently, there has been a large number of erroneous citations to our original work. The objective here was to conduct a bibliometric analysis to quantitate the accuracy of citations to Murphy and Westmark’s publication entitled, “Folic Acid Fortification and Neural Tube Defect Risk: Analysis of the Food Fortification Initiative Dataset”. We found a 70% inaccuracy rate. These findings highlight the dire need for increased rigor in citing scientific literature, particularly in regard to biomedical research that directly impacts public health policy.

## 1. Introduction

Neural tube defects (NTDs) are serious birth defects that affect the brain and spinal cord and can result in infant mortality or serious disability. The prevalence of NTDs is in the range of 4–32 per 10,000 births with a mean of 13 [1], and 0.3–199.4 per 10,000 births in another study [2]. Adequate folate in the diet during the periconceptual period may reduce the incidence of NTDs by 50% [3]. In 1998, the United States of America (U.S.A.) was the first country to mandate national fortification of cereal grains with folic acid to reduce NTDs. As of November of 2023, 94 countries have mandatory cereal grain fortification legislation [4]. It is estimated that 97% of industrial-scale wheat flour was fortified in the Americas, 31% in Africa, 44% in the Eastern Mediterranean, 21% in Southeast Asia, 6% in Europe, and 4% in Western Pacific regions in 2007 [4,5]. While substantial evidence supports the role of folate in preventing NTDs, the efficacy of fortification of food products with folic acid is debated. Folic acid, which is a synthetic form of folate used for food fortification and vitamin supplements, has a different chemical structure than natural folates found in fruits and vegetables and needs to be converted into 5-methylenetetrahydrofolate, the major circulating form of folate for use by the body as vitamin B9.

In 2020, we published a study in *Nutrients* comparing NTD prevalence in countries with versus without national folic acid fortification policies using the publicly available Food Fortification Initiative (FFI) dataset [1]. The FFI is a global group that provides technical assistance to governments regarding food fortification to address micronutrient deficiencies. The rationale was that fortification would directly affect the majority of inhabitants in a country. The null hypothesis was that there would be no association between national folic acid fortification and the prevalence of NTDs. The alternative hypothesis was that national folic acid fortification alters the prevalence of NTDs. The primary endpoint of interest was the prevalence of NTDs. The primary predictor variable was folic acid fortification. The average prevalence of NTDs per 10,000 births was calculated as a function of folic acid fortification after binning countries into groups based on which cereal grain was fortified. We found an equivalent average as well as a range of values for NTDs per 10,000 births with and without fortification. We also stratified the FFI NTD data based on national economic indicators and found a strong linear correlation between higher socioeconomic status (SES) and reduced NTDs. Others have also shown correlations between better SES and reduced prevalence of NTDs [2,6]. In the absence of prospective monitoring, it is not possible to determine cause and effect; however, the cumulative data strongly suggest that higher SES, not fortification of cereal grains with folic acid, contributes to reduced prevalence of NTDs.

Food fortification is a major public health issue that has been surrounded by controversy. Kancherla et al. disagreed with the statistical analysis and conclusions of our paper, which are discussed in detail in [7,8]. In our reply [8], we pointed out that a strong case can be made that proponents of national food fortification commit exception fallacies in their analyses. Exception fallacies occur when conclusions about a group of people are based on data from individual cases. In 2022, in the U.S.A., the number of women of childbearing age, 15–44, was 65 million out of a total of 333 million people, which is 20% of the total population. Not all women and their babies are deficient in folate and not all will benefit, and some may actually be harmed, from excess consumption of folic acid dependent on dietary patterns and genetics. Folic acid can interact with medications as well as increase the risk of certain types of cancer including prostate [9,10]. The Cochrane Database of Systematic Reviews assessed the efficacy of folic acid fortification on health outcomes in the overall population as “low certainty” in improving NTD outcomes [11].

The concern addressed in this paper is the high incidence of inaccurate citations regarding our 2020 study. The research process is dependent on the accurate citation of prior scientific findings. Examples of citation inaccuracies include (1) selective citation, or choosing references in an arbitrary fashion; (2) citation bias, which is the preferential citing of positive findings; (3) secondary citation, or not citing the original source; (4) incorrect/opposite conclusion, which occurs when inaccurate or missing information is cited, for example, citing an article presenting the opposite conclusion referred to in the study; and (5) fact not found, which occurs when an cited article does not mention the cited claim [12]. Improper and incomplete referencing of the prior related literature impedes biomedical research and can substantiate poor public health policies. The objective here was to conduct a bibliometric analysis to quantitate the accuracy of citations to Murphy and Westmark’s publication entitled, “Folic Acid Fortification and Neural Tube Defect Risk: Analysis of the Food Fortification Initiative Dataset” [1].

## 2. Materials and Methods

### 2.1. Study Design, Search Strategy, Inclusion Criteria, and Data Collection

The study design was a content analysis of citations to Murphy and Westmark, 2020 [1]. The search strategy involved retrieval of all citations from Google Scholar at scholar.google.com from the time of publication, 18 January 2020, to 30 May 2024, using the title of the paper. There was a total of 52 citations [6,7,8,13,14,15,16,17,18,19,20,21,22,23,24,25,26,27,28,29,30,31,32,33,34,35,36,37,38,39,40,41,42,43,44,45,46,47,48,49,50,51,52,53,54,55,56,57,58,59,60]. Inclusion criteria included available full texts. One citation was excluded because it was a pre-print and the published text was available (i.e., duplicate publication) [50]. One citation was a book chapter and was excluded because the University of Wisconsin-Madison Library could not retrieve a copy for review [60]. Google Translate was employed to interpret manuscripts in non-English languages. 

### 2.2. Literature Analysis and Statistics

The citing articles were arranged in alphabetical order by the first author’s last name in Table 1. The extracted data included: the name of the first author, institution, country, year work was published, name of the journal, title of the paper, and copy of the text citing Murphy & Westmark, 2020 in quotes. Notes were added after the quoted material to clarify discrepancies between the citations and the cited work. Both authors reviewed and graded the 50 included texts for citation accuracy. Inaccurate citations were further reviewed to determine the type of error (selective citation, citation bias, secondary citation, incorrect/opposite finding, and fact not found), where secondary citations were considered less severe and other types of errors were labeled as serious. The entries in Table 1 were color-coded and labeled with the type of error per the figure legend. The data were analyzed in accordance with STROBE guidelines and the BIBLIO checklist [61,62]. A copy of the BIBLIO checklist is provided in Appendix A. Percentages were computed to describe the results. To statistically test for differences in accurate citation rates, Fisher Exact tests were used and *p* values were reported. Statistical significance was defined as *p* < 0.05.

## 3. Results

### 3.1. Citation Inaccuracies in Murphy & Westmark, 2020

The paper, “Folic Acid Fortification and Neural Tube Defect Risk: Analysis of the Food Fortification Initiative Dataset” by Murphy and Westmark, averaged 1 citation per month since its publication in January 2020 (52 citations). Here, we found a 70% inaccuracy rate in citing this work (Table 1), which is statistically significantly higher than an inaccuracy of 12.3% for all citations from 19 other papers published by the same laboratory in the same timeframe [130 non-duplicated full-text publications available: 88% correct; 2.3% mis-citations; 10% secondary citation errors]; *p* < 0.0001 [8,14,63,64,65,66,67,68,69,70,71,72,73,74,75,76,77,78,79]. Others have reported citation inaccuracy at 25% in medical journals including half major and half minor/secondary citation errors [80]. The serious citation errors in Murphy and Westmark, 2020 [1], included 18 papers with selective citation errors, 12 papers with citation bias errors, 21 papers with incorrect/opposite finding errors, and 6 papers with fact not found errors within 25 publications, i.e., half of the papers evaluated contained one or more major errors. Secondary citation errors were found in 15 publications. The errors in 19 other laboratory publications were predominantly secondary citation errors. This bibliometric analysis only includes publications that cited Murphy and Westmark, 2020 [1]. We did not perform a comprehensive analysis of the 294 citations, between 18 January 2020 and 30 May 2024, in PubMed with the search terms “folic acid AND fortification” to determine errors in selective citation where the Murphy and Westmark, 2020 paper should have been cited to discuss alternate findings but was not. Of interest, publications from countries with mandatory legislation regarding the fortification of wheat, as of November 2023 (FFI website [4]), had an increased number of papers with serious citation errors, i.e., 64% of the papers in which the corresponding author was from a country with mandatory fortification had serious errors, whereas 77% of papers from authors in countries without mandatory fortification cited Murphy and Westmark, 2020 correctly, *p* = 0.038 (note self-citations excluded from analysis).

### 3.2. Trainee Theses and Dissertations Have a High Error Rate

Seven of the citations were from theses or dissertations, and all contained inaccuracies (five serious errors and two secondary citation errors). Removing the trainee theses and dissertations from the citation pool resulted in 43 papers with a 65% overall error rate and a 47% serious error rate in citations, which is not statistically different from the trainee theses and dissertations with an overall error rate of 100% (*p* = 0.087) and a serious error rate of 71% (*p* = 0.42).

## 4. Discussion

These findings highlight the dire need for increased rigor in citing the scientific literature, particularly in regard to biomedical research that directly impacts public health policy. NIH promotes increased rigor and reproducibility in biomedical research through multiple initiatives. Per their website, “Two of the cornerstones of science advancement are rigor in designing and performing scientific research and the ability to reproduce biomedical research findings” [81]. The National Library of Medicine lists Research Reporting Guidelines and Initiatives by Organizations that provide advice from 50 organizations for reporting research methods and findings [82]. Advice is provided on areas including formatting style (AMA Manual of Style and APA Style), animal metrics (ARRIVE), clinical trials (ASSERT, CONSORT), case reports (CARE), systematic reviews and meta-analyses (PRISMA), observational research (STROBE), etc. Per NIH Principles and Guidelines for Reporting Preclinical Research, transparency in reporting should include standards, replicates, statistics, randomization, blinding, sample size estimation, and inclusion and exclusion criteria. A clear and transparent account of what and how research was performed is imperative to reproduce findings and assess the rigor of a study. We assert that a third cornerstone, i.e., accurate citation of the prior literature, is also paramount to rigorous biomedical research, which forms the foundation for public health policy.

Our bibliometric analysis found a 70% inaccuracy rate in citing Murphy and Westmark, 2020, with half of the papers containing serious citation errors. It is perplexing that numerous authors chose to inaccurately cite rather than discuss work that contradicted their findings. It is also disturbing that seven of the citations were from trainee theses or dissertations, including 71% with serious errors and the remaining 29% with secondary citation errors. With the limited number of trainee publications, the error rate approached but was not statistically different from the other publications (*p* = 0.087) There is a dearth of formal training in scientific writing in graduate school programs. Training may be focused on format and style, not content or accurate reflection of the findings. Students typically learn to write and cite scientific research papers after reading and evaluating the primary literature and through mentor feedback. A 100% error rate is of concern in regard to the quality of literature analysis skills and mentor oversight of writing in graduate training programs. Solid scientific writing requires more than good grammar, appropriate word choice, and telling a coherent story; it requires rigorous analysis and presentation of the prior literature.

Appropriate citation of the prior literature is imperative for research, and, in particular, for science informing public health issues. While our ecological study on population-level efficacy of food fortification with folic acid did not show a benefit [1], and we are unaware of any appropriately controlled population-level study showing efficacy, the biology of folic acid has been extensively studied and there is ample evidence of potential adverse effects [83,84]. Numerous reports indicating a decline in the prevalence of NTDs with mandatory folic acid fortification do not consider that NTDs were declining without fortification [85,86] and do not include a comparison with a non-fortification group during the same time period.

Inaccurate citation, particularly through opposite conclusions, buries science that needs to be debated and impedes the discovery and implementation of efficacious public health policies. For example, public health measures targeting at-risk populations to increase education on adequate nutrition in the pre-conception period as well as access to healthy foods could reduce NTDs. In the Herter-Aeberli study, only 25% of women received information from their gynecologist about folate deficiency and the risk of birth defects before they became pregnant [24]. Only 7% of the women received information about nutrition prior to their pregnancy [24]. Higher SES status exerts a significant effect on folic acid status and NTD risk. We found a strong linear relationship between reduced NTDs and higher socioeconomic status (SES) [1]. Herter-Aeberli and colleagues demonstrated that education was an important predictor of the timing during pregnancy when women took supplements [24]. Specifically, post-secondary education was associated with an increased use of supplements prior to pregnancy, i.e., 41% versus 16% of women without a post-secondary degree [24]. Likewise, higher income was associated with recommended supplement use, i.e., 43% versus 29% of women in the lower income class [24]. Others show that neighborhood deprivation, i.e., lower SES, is associated with increased NTDs [6,87].

A personalized medicine approach in regard to folate and folic acid treatment is needed. There are several vulnerable populations that may be adversely affected by over-exposure to folic acid, such as the elderly who have low vitamin B12 levels, those taking medications that interact with folic acid, and those with certain methylenetetrahydrofolate reductase (MTHFR) polymorphisms. There is potential for nervous system damage in persons with low or insufficient B12 who are exposed to a high intake of folic acid [88,89]. For example, low vitamin B12 in the elderly in conjunction with higher serum folate is associated with anemia and cognitive impairment [90,91,92,93]. Folic acid interacts with 35 medications including chemotherapy, seizure, and cholesterol drugs [10]. Folic acid also interacts with genetic mutations, for example, MTHFR, which is a critical enzyme in folate metabolism affecting DNA methylation, synthesis, and repair.

It is worth discussing a potential mechanism through which folic acid could affect a substantial portion of the population. The cytosine-to-thymine substitution at base pair 677 (C677T) of the MTHFR gene results in the substitution of a valine codon for the evolutionarily conserved alanine residue. This genetic variant is common in North America, where approximately 12% of individuals are homozygous for the C677T allele and 40–45% are heterozygous [94,95]. Persons with the C677T polymorphism exhibit decreased enzymatic activity of MTHFR, which is the final enzyme in a multi-step pathway that converts folic acid to the active folate metabolite 5-methyltetrahydrofolate (5-MTHF), the form that can be transported into cells and across the blood–brain barrier. With the decreased activity of MTHFR(C577T), the final conversion step of folic acid to 5-MTHF may not be complete, leading to high levels of unmetabolized folic acid (UMFA) and increased homocysteine levels.

UMFA was detected in 38% of older adults in the National Health and Nutrition Examination Survey (NHANES) U.S.A. study population after fasting [96]. A double-blind, randomized controlled trial in Australia found more UMFA in serum from women consuming an 800 μg/day folic acid supplement from 12–16 weeks gestation until 36 weeks of gestation compared with the control group [50]. Moderately high folic acid intake has negative effects on embryonic development in mice with an increased prevalence of embryo loss and defects in heart development [97]. In adult rats, folic acid promotes spinal axon regeneration albeit in a bell-shaped dose-dependent and methylation-dependent manner with effects that may persist over several generations [98]. Possible repercussions of excessive folic acid consumption are increased UMFA levels, competition with natural folate for binding to folate receptors, downregulation of intestinal and renal folate uptake, and altered epigenetic programming, which has been postulated to contribute to autism prevalence, NTDs, and sperm degradation. [99,100,101,102,103]. In postmenopausal women with higher RBC folate, altered epigenetic programming in the form of lower DNA methylation was observed in the post-folic acid fortification period [104]. We currently lack clinical data regarding UMFA and neuronal tube defects. Using the PubMed search terms “unmetabolized folic acid” AND “clinical trial” returns 21 publications, but none assess NTDs.

The rates of autism and Alzheimer’s disease have increased substantially since the late 1990s when folic acid fortification was mandated in the U.S.A. [105]. It has been hypothesized that enhanced maternal/fetal folate status during pregnancy alters natural selection for autism [106]. Folate stimulates the expression of secretases involved in the non-amyloidogenic processing of amyloid protein precursor (APP) (ADAM9 and ADAM10) while inhibiting the expression of the amyloidogenic secretase BACE1 [107]. Furthermore, hyperhomocysteinemia increases amyloid-beta levels [108]. These data suggest that excess folic acid could interfere with APP processing pathways, which are implicated in Alzheimer’s disease, fragile X syndrome (FXS), autism, and other neurological disorders [109,110,111]. Meta-analyses indicate a significant association between the MTHFR C677T polymorphism and the risk of NTD pregnancy, autism, and Alzheimer’s disease [112,113,114]. Folic acid deficiency is associated with diverse neuropsychiatric symptoms, yet treatment with folic acid increases psychotic behavior in psychiatric patients [115]. Thus, there is a need to determine optimal folate dosing for the prevention of NTDs while avoiding other neurological disorders and cancer.

It is past time to reconsider public health policy regarding fortification of food with folic acid. There is a lack of prospective monitoring of adverse health effects associated with the consumption of folic acid and concerns surrounding exceeding the upper tolerable limit. Mandatory folic acid fortification in the U.S.A. increased average intake by twice what was projected [116]. Despite mandatory fortification of cereal grains with folic acid in the U.S.A., the estimated prevalence of folate insufficiency based on red blood cell (RBC) folate concentration in women of reproductive age was 23% for NHANES 2007–2012 participants [117,118]. Thus, a large portion of the target population is deficient in folic acid despite fortification and a large portion is at risk for adverse health events in response to fortification.

The Code of Federal Regulations (CFR)—Title 21—Food and Drugs §101.79 states the following regarding health claims associated with folate and NTDs, “The available data show that diets adequate in folate may reduce the risk of neural tube defects. The strongest evidence for this relationship comes from an intervention study by the Medical Research Council of the United Kingdom that showed that women at risk of recurrence of a neural tube defect pregnancy who consumed a supplement containing 4 milligrams (mg)(4000 micrograms (mcg)) folic acid daily before conception and continuing into early pregnancy had a reduced risk of having a child with a neural tube defect. (Products containing this level of folic acid are drugs). In addition, based on its review of a Hungarian intervention trial that reported periconceptional use of a multivitamin and multimineral preparation containing 800 mcg (0.8 mg) of folic acid, and its review of the observational studies that reported periconceptional use of multivitamins containing 0 to 1000 mcg of folic acid, the Food and Drug Administration concluded that most of these studies had results consistent with the conclusion that folate, at levels attainable in usual diets, may reduce the risk of neural tube defects”. Please note that these regulations from the FDA: (1) cite the benefits of supplements not fortification; (2) regard folic acid supplementation studied in at-risk populations, i.e., women at risk for a second NTD-pregnancy; (3) state that products containing high levels of folic acid are considered drugs; dose is not known with fortification as individuals are exposed to varied quantities and types of food; (4) state that supplementation occurred daily before and during early pregnancy, not lifelong; and (5) infer that folate levels adequate to reduce NTDs can be obtained from a typical diet.

Recommendations should also consider potential interactions with agrochemicals. National fortification of cereal grains with folic acid commenced in 1998 at the same time genetically modified foods were introduced into the food supply in the U.S.A. There is a strong association between the increased use of glyphosate (the active ingredient in RoundUp Ready^®^) on genetically modified soy and corn crops and the prevalence of numerous medical conditions including autism [105]. The gut bacteria that metabolize folate are strains of *Lactobacilli* and *Bifidobacteria* [119], which are microbes that are preferentially killed by glyphosate [120,121].

Regarding study limitations, a confounding issue of our original analysis demonstrating a reduced prevalence of NTDs with higher SES is that there could be increased maternal use of folic acid supplements during pre-conception and pregnancy with higher education level and increased health knoweldge. The strength of this study is the transparent, quantitative approach to assessing citation accuracy.

## 5. Conclusions

In conclusion, accurate citation of scientific findings needs to be included with robust and unbiased experimental design, methodology, analysis, interpretation, and reporting of results as rigor variables in the conduct of science. Particularly troubling are opposite conclusion errors in citation where the opposing data should be discussed instead of misrepresented. The lack of rigor in accurately citing the scientific literature impedes the promotion and implementation of effective public health policies. An honest and critical evaluation of the folic acid fortification literature and admittance of shortcomings in efficacy evaluation is long overdue. National fortification policies were instituted to prevent NTDs during pregnancy; however, the potential effects on miscarriages, autism, Alzheimer’s disease, and cancer would warrant a “first, do no harm” approach. In the absence of a monitoring program at the individual subject level, it is impossible to discern the full benefit or consequences of national folic acid fortification policies. Many individuals are at risk for exceeding the safe upper limit, and many are at risk for deficiency because of food choices and availability. A failure to evaluate the health effects of fortification prospectively will negatively impact vulnerable populations. Overall, we have not witnessed the projected 50% reduction in NTDs in the U.S.A. in response to the national fortification of cereal grains with folic acid, and current fortification levels present health concerns for large sub-groups of the general population. A one-size-fits-all strategy to medicine, and food is medicine, is not working. There is an urgent need for a personalized medicine approach with regard to folic acid.

## Figures and Tables

**Table 1 nutrients-16-02503-t001:** Bibliometric Analysis of Murphy and Westmark, 2020. * Number and color scheme for citation scores: 0—no error, 1—selective citation error, 2—citation bias error, 3—secondary citation error, 4—incorrect/opposite conclusion error, 5—fact not found error, and ?—probable typographical error. Green = correct citation, blue = correct self-citation, yellow = secondary citation error, and red = serious citation error. Serious citation errors included selective citation, citation bias, incorrect/opposite conclusion and fact not found. Citations within quotes in the Text Citing Murphy & Westmark, 2020 column were removed to avoid confusion with references in the paper.

First Author	Institution	Country	Year	Journal	Paper Title	Text Citing Murphy & Westmark, 2020	Error *
Abu Salem [48]	American University of Beirut	Lebanon	2024	Preprint	Congenital anomalies in the clinic and ambient air pollution associations: evidence from responsible artificial intelligence	“Such programs also help guide planning and prevention and assess the effectiveness of a specific intervention at the public health level”.	0
Al-Wassia [13]	King Abdulaziz University	Saudi Arabia	2023	*Med Arch*	Neural tube defects from antenatal diagnosis to discharge—a tertiary academic centre experience	“Folic acid supplementation and food fortification decreased the occurrence of NTDs by 30–50% in many countries”. Note: Murphy and Westmark, 2020 data does not show decreased NTDs in response to fortification.	4
Alam [14]	Oklahoma State University	United States	2023	*Frontiers in Neuroscience*	Diet in treatment of autism spectrum disorders	“A personalized medicine approach will be required to implement micronutrient-mediated therapy for ASDs, particularly considering the contradictory evidence regarding the efficacy of population level folic acid supplementation and neural tube defects”.	0
Alzergy [15]	Omar Al-Mukhtar University	Libya	2020	*IOSR Journal of Pharmacy and Biological Sciences*	Effect of folic acid supplementation in early pregnancy on spinal cord in mice embryos	“The United States implemented mandatory fortification of FA in 1998 to prevent neural tube defects during pregnancy. The health benefits of folic acid are well documented however, there are potential risks of exceeding the upper tolerable limit. Murphy and Westmark also, concluded that national fortification with folic acid is not associated with a significant decrease in the prevalence of neural tube defects at the population level”.	0
Baqar [16]	University of Ottawa	Canada	2021	Thesis	Neural tube defects and co-occurring anomalies in Europe, 1980–2015	Note: this thesis regards national folic acid fortification for Europe. It listed the Murphy and Westmark, 2020 paper in the bibliography but there was no mention or discussion of the data in the text, which was directly relevant but contradicted statements in the thesis.	1,4
Barriga [17]	Servicio de Genética & EIM, Instituto Nacional de Salud del Niño-Breña	Peru	2021	*Rev Neuropsiquiatr*	Genomic advances in the last decade and their influence on the diagnostic approach to intellectual disability.	“Other relatively common teratogens include isotretinoin. (induces abnormal expression of genes involved in neuronal differentiation), misoprostol, folic acid deficiency and rubella infection”.	3
Bellanger [18]	University of the Incarnate Word, Feik School of Pharmacy	United States	2021	*Side Effects of Drugs Annual*	Complementary and Alternative Medicine	“A global focus on quality food as a requirement for healthy communities can provide incentives for the food industry interests to produce higher nutrient content at a lower cost. The policy in many countries is to supplement micronutrients, vitamins, and minerals in staple foods, such as, grain, dairy, and other products. This supplementation has reduced folic acid, iodine, iron, and vitamin deficiencies, and has been responsible for better-targeted health outcomes in specific populations. However, these policies have the potential to increase risks from overdoses in some vulnerable populations”. Note: we discussed and cited vulnerable population data but did not conduct the primary research.	3
de la Fournière [21]	Sorbonne University	France	2020	*Nutrients*	Prevention of neural tube defects by folic acid supplementation: a national population-based study	“The socioeconomic status has been reported as strongly associated with NTDs prevalence, thus questioning the actual contribution of food folate fortification in several countries. However, mandatory fortification can be considered one of the most effective and cost effective public health measures to date: the estimated annual economic gain in the USA is 603 million dollars”. Note: SES data are cited correctly, but their conclusion disagrees with Murphy and Westmark, 2020 findings showing poor correlation between folic acid fortification and NTD prevalence, which they do not discuss.	1,2,4
De Melo [20]	Federal University of Rio Grande do Norte	Brazil	2023	*Food Chemistry*	Biological and pharmacological aspects of tannins and potential biotechnological applications	“This action may prevent the subsequent translocation of luminal antigens via the mucosa to the whole body, which may change intestinal mucosal homeostasis, consequently increasing susceptibility to systemic and chronic inflammation and changing nutrient absorption”.	5
Elahi [19]	Hormozgan University of Medical Sciences	Iran	2023	*Research Square Pre-print*	Maternal smoking during pregnancy and its effects on neural tube defects: a holistic systematic review	“What has attracted the attention of the scientific community is the prevention and prognosis of these disorders, which implies the need to know the risk factors of its occurrence. Normal body mass index, consumption of nutritional supplements such as folic acid or multivitamins, proper health habits of the mother, presence of dangerous teratogens such as alcohol consumption or smoking, folic acid deficiency, family marriage and genetic issues are the main causes of NTD in pregnant mothers”.	3
French [22]	DSM Nutritional Products	Japan	2023	*Evidence-based Functional Foods for Prevention of Age-related Diseases*	Chapter 9 Micronutrient status among adults in the Asia Pacific and potential impact on age-related diseases	“Mandatory folic acid fortification programs in the USA, Canada, Costa Rica, Chile, and South Africa are associated with significant increases in blood folate concentrations and declines of 25–50% in the prevalence of neural tube defect (NTD) affected pregnancies”. Note: we cited this statement at the beginning of the Discussion [1], and then discussed confounding issues and contradictory data, which disagree with the conclusion.	1,2,3,4
Harper [23]	University of Colorado School of Medicine	United States	2023	*Fertility and Sterility*	Optimizing preconception care and interventions for improved population health	“Folic acid supplementation through a daily prenatal vitamin starting before conception remains one of the most simple and effective preconception interventions available to date. The United States was the first country to mandate a national folic acid food fortification program to prevent neural tube defects including spina bifida. More than 80 other countries have followed suit in fortifying cereal grains with folic acid”. Note: Murphy and Westmark, 2020 data [1] are not discussed in relation to contradictory statements in the paper: “Additionally, public health initiatives to fortify grain products with folic acid have been undertaken to allow for broader optimization of folic acid levels among individuals of reproductive age. These strategies have drastically reduced the rate of neural tube defects in developed countries, with the residual risk among individuals with optimal folic acid stores decreasing from 25 to 5 in every 10,000 births”. “Public health measures to promote preconception health practices and policies (e.g., folic acid fortification of food supply, …) can lead to more favorable pregnancy outcomes.	1,2,4
Herter-Aeberli [24]	Department of Health Sciences and Technology	Switzerland	2020	*Nutrients*	Inadequate status and low awareness of folate in Switzerland-a call to strengthen public health measures to ensure sufficient intakes	“It is estimated, that mandatory folic acid fortification has prevented >1300 NTDs every year in the US, yet the practice is still highly debated. Concerns involve potential risk of exceeding the upper tolerable intake limit in vulnerable groups, possibly increasing the risk for: (1) unmetabolized folic acid; (2) potential epigenetic changes and increased cancer risk, and; (3) masking of diagnostic signs of vitamin B12 deficiency”. Note: Citation to Murphy and Westmark, 2020 [1] is correct regarding “highly debated”, but the citation regarding concerns/risks is a secondary citation. The paper did not discuss the lack of efficacy Murphy and Westmark, 2020 [1] found for folic acid fortification, yet promoted mandatory fortification of staple foods, which is a selective citation error. The statement, “Mandatory fortification of staple foods with folic acid is another proven intervention to increase the folate intake in the general population and decrease the incidence of NTDs” is an opposite finding error.	1,3,4
Heydari [25]	Research Institute of Dental Sciences, Shahid Beheshti University of Medical Sciences	Iran	2024	*BMC Oral Health*	Prevalence, trend, and associated risk factors for cleft lip with/without cleft palate: a national study on live births from 2016 to 2021	“Consumption of folic acid before and during early pregnancy reduces the chance of neural tube defects and oral clefts”. Note: This could be considered a type 5 error-fact not found as we did not show reduced NTD with folic acid fortification. Alternatively, it could be considered a secondary citation error as perinatal folic acid supplementation has been shown effective by others. Regardless, the paper cites but does not discuss our data, which is a type 1 error.	1,3,5
Huang [26]	Tianjin Medical University	China	2022	*Aging & Mental Health*	Baseline folic acid status affects the effectiveness of folic acid supplements in cognitively relevant outcomes in older adults: a systematic review	“Currently, more than 80 countries or regions have used folic acid-fortified cereals, while other countries, including China, have not yet implemented national folic acid-fortified foods”.	0
Isaković [27]	Omnion Research International, University of Zagreb	Croatia	2022	*Biomedicines*	Overview of neural tube defects: gene-environment interactions, preventative approaches and future perspectives	“Regardless of the multitude of reported benefits, some studies reevaluating the mandatory FA fortification procedure are still present. One of the most prominent ones includes a population-based analysis of the dataset collected by the Food Fortification Initiative (FFI, Atlanta, GA, USA), a multi-sector partnership that works to reduce micronutrient deficiencies—including folic acid and iron—through the promotion of flour and rice fortification and aids with the planning and implementation of fortification programs. The subsequent analysis of the aforementioned dataset suggested that there exists only a weak correlation between NTD prevalence and the level of FA food fortification. Rather, the authors indicated that they observed a strong linear relationship between reduced NTDs and better socioeconomic status (SES). However, because Murphy et al. performed a population-level observational retrospective study, there exist several confounding factors potentially impacting the significance of their conclusion. As such, the overarching consensus still remains, advocating for FA supplementation. Nevertheless, as more data become available that could facilitate long-term studies, additional research is needed to further evaluate all the hypotheses and address their impact on public health. Even though the health benefits of FA supplementation are well documented, risks associated with food fortification, potentially leading to high levels of unmetabolized folic acid (UMFA), are still being discussed”.	0
Ismail [28]	Qatar University	Qatar	2023	*Foods*	Intended and unintended benefits of folic acid fortification—a narrative review	“The USA was the first nation to implement mandatory folic acid fortification for the prevention of NTDs, which has significantly reduced their incidence. Currently, more than 80 other countries have adopted the US policy of fortifying cereal grains with folic acid”. Note: The first reference is a secondary citation error. The serious errors concern the major conclusion of the paper, “The current implementation of mandatory folic acid fortification has achieved its intended benefit of reducing the incidence of NTDs in newborns”, which contradicts the findings of Murphy and Westmark, 2020, which are not discussed. These are selective and opposite finding errors.	1,2,3,4
Jaffe [29]	University of Colorado	United States	2021	Dissertation	Elucidating the role of grainyhead-like 3 in the regulation of cellular projections during murine neural tube closure.	“Regions with lowest NTD rates globally include the two Americas (North America: 7.5 NTD per 10,000 births; Latin America 8.4 per 10,000) and Europe (9.7 per 10,000), while those with the highest rates include Northern Africa (17.4 per 10,000), East Asia (19.4 per 10,000) and Southern Asia (31 per 10,000). These differences may reflect general socio-economic or dietary and nutritional disparities between regions, as reflected by differences in NTD rates between regions with higher median income status (Europe and North America) and lower median income status (Sub-Saharan Africa; South Asia)”. "Folic acid (FA) fortification has been roundly successful worldwide in decreasing NTD incidence at a population level, with most countries exhibiting a 30–60% decline in NTD rate in the years following fortification. Variability in the size of the effect that folic acid supplementation has on NTD rate is likely reflective of differences in starting NTD rates due to other factors like genetic background and overall socio-economic status of the population in a given region”. “Socio-economic status remains one of biggest correlates to NTD risk”. Note: This dissertation selectively cited our data that agree with their premise without discussing data that disagree.	2,4
Jankovic-Karasoulos [30]	Flinders University	Australia	2023	*Nutrients*	Elevated maternal folate status and changes in maternal prolactin, placental lactogen and placental growth hormone following folic acid food fortification: evidence from two prospective pregnancy cohorts	“To ensure that all women of reproductive age have sufficient folate levels, including those who may conceive unintentionally and are unlikely to supplement, over 80 governments across the world implemented mandatory FA food fortification programs”. Note: The paper also states, “Despite the effectiveness of FA food fortification, …”, but does not discuss our findings that show lack of efficacy.	1,2,4
Kancherla [7]	Emory University Rollins School of Public Health	United States	2020	*Nutrients*	The fallacy of using administrative data in assessing the effectiveness of food fortification. Comment on: "Folic acid fortification and neural tube defect risk: analysis of the food fortification initiative dataset. Nutrients 2020, 12, 247”	Note: their whole paper is in response to the Murphy and Westmark, 2020 publication using the FFI dataset. There are numerous incorrect statements regarding the data, which were clarified in our response to the editor [8]. The authors do provide a review of potential shortcomings, which were either addressed in the original publication [1] or the response to the editor [8] or cannot be addressed with available data.	4
Kishida [31]	University of Tsukuba	Japan	2024	*Nutritional Neuroscience*	Serum folate and risk of disabling dementia: a community-based nested case-control study	“…short-term Korean prospective studies could make it difficult to examine blood folate and dementia associations properly because of the long latent period of dementia development. We extended the evidence from these Korean studies by our longterm follow-up (mean 20.8 years) and provided the evidence from countries where fortification of flour was not mandatory, unlike in the United States and Canada”.	3
Kuhn-Gale [32]	University of Colorado, Boulder	United States	2021	Thesis	Multigenerational folic acid supplementation disrupts HMGB1 nuclear localization through decreased reactive oxygen species	“To increase the number of women consuming sufficient folic acid, most developed countries fortify some or all of their grains with folic acid, yet a mechanism for this protective effect is still not understood4”. Note: Murphy and Westmark, 2020 data [1] do not show a protective effect.	4
Lawler [33]	University of Wisconsin-Madison	United States	2023	*Nutrition and Cancer*	Associations between folate and alcohol consumption and colorectal tumor Ki67 expression in the Southern Community cohort study	“The potential dual effect of folate is concerning given mandatory folic acid fortification in much of the world,…”	0
Leng [34]	University of Otago	New Zealand	2020	Thesis	Dietary folate intake, sources and adequacy of New Zealand adolescent males	“Folate can also be found in food supply by means of folic acid fortification via regulations enforced either voluntary, mandatory or both. Regulations for mandatory fortification of food vehicles such as flour are currently in place in over 80 countries to date”. “The mean folate intake for 295 males aged 12.5–17.5 years was 215.1 μg folate/d. However, it is unclear whether the folate intake is in DFE albeit most European countries has minimal voluntary fortification and none of these countries have mandatory folate fortification for their wheat, maize or rice flour as of April 2020”. Note: This thesis did not discuss findings from Murphy and Westmark, 2020 that were directly relevant to their study while stating: “Fortification has been introduced in many countries worldwide, with mandatory fortification programmes proving efficacious at meeting the folate requirement of the whole population…” and with the objective of “providing updated information on dietary intakes to aid future dietary recommendations and policy”.	1,2,4
Li [57]	The Second Hospital of Shanxi Medical University, Taiyuan	China	2021	*J Int Obstet Gynecol*	Research progress of folic acid and its metabolism in cervical cancer	“However, some scholars hold opposing views, believing that excessive folic acid intake may have a counter-effect: such as masking vitamin B12 deficiency, reducing the activity of natural killer cells in elderly women, and promoting the occurrence of chronic diseases and certain cancers”.	3
Ling [35]	Jinan University First Affiliated Hospital	China	2024	*The Journals of Gerontology*	Associations of folate/folic acid supplementation alone and in combination with other B vitamins on dementia risk and brain structure: evidence from 466 224 UK Biobank participants	“Consequently, some nations have implemented compulsory folic acid fortification programs. However, several Western European countries have yet to enforce mandatory folic acid fortification, partly due to concerns about potential adverse effects on cancer incidence. Furthermore, a recent data analysis examining the correlation between national folic acid fortification, and the risk of NTDs found no significant association between folic acid fortification and a decreased incidence of NTDs in the population. Therefore, the potential negative consequences of folic acid supplementation should be a focal point of attention”.	0
Lucock [36]	University of Newcastle	Australia	2022	*Exploratory Research and Hypothesis in Medicine*	A brief introduction to Darwinian medicine	“It is fortuitous that this evolutionary mismatch, which led to serious morbidity and mortality, can be corrected by the simple measure of introducing folate into the diet through both mandatory and discretionary fortification programs. This is a good example of DM, with the bonus of preventing neural tube defect-affected pregnancy, 21 which is ostensibly the main reason for fortification. In order to provide context, in 1995 it was calculated that mandatory folic acid fortification in the US could prevent approximately 50,000 coronary artery disease deaths each year. Three years later, in 1998, mandatory fortification was implemented. Today, at least 80 countries have adopted a similar fortification strategy”. Note: The paper supports mandatory fortification without discussing contradictory findings in Murphy and Westmark, 2020.	1,2,4
Lugo [37]	Villa Clara University of Medical Sciences, Mariana Grajales University	Cuba	2022	Dissertation	Genetic and environmental factors in mothers with offspring affected by folate-sensitive congenital defects in Villa Clara	“Folate is an essential nutrient, because mammals lack the physiological activity to synthesize it, it is necessary to obtain it from the diet or consume its synthetic form: FA. There is sufficient epidemiological evidence to support the fact that maternal preconception supplementation with FA reduces the appearance of NTD and other folate-sensitive DC in the offspring”.	3
Madle [38]	Univerzita Karlova	Czech Republic	2021	Thesis	The importance and need of folic acid during pregnancy.	“A number of positive effects, especially in connection with the prevention of congenital defects in the area of the neural tube, are attributed to a sufficient intake of folic acid before pregnancy and in the first weeks of pregnancy. The positive effects of increased folic acid intake, and not only for the prevention of congenital developmental defects, have been studied worldwide since the 1960s”. “If foods containing folic acid are exposed for some time to even ordinary daylight (especially vegetables), sunlight, long storage, let alone heat treatment, its content decreases significantly. When food is heat-treated, losses of most folates in food tend to be more than 50%, with leafy vegetables the loss can even be as high as 95%”. “It is therefore a matter of the legally mandated fortification of foods with folic acid, mostly directly of the basic ones, i.e. flour (wheat and corn) and rice. In some countries, even the basic types of bakery products, mostly bread, as well as some cereals, cereals, and pasta, are fortified compulsorily”. “According to statistics, this mandatory fortification soon proved to be very effective in the fight against the occurrence of birth defects, especially in connection with the reduction of NTDs, i.e. mainly spina bifida, anencephaly, encephalocele”. “These studies also came out clearly in favor of fortification and a subsequent significant reduction in the incidence of NTDs”. “Studies from recent years try to point to the fact that mandatory fortification of foods with folic acid is not associated with a significant decrease in the prevalence of NTDs at the population level”. “Also due to the fact that neither in the EU, nor in the Czech Republic, unlike other countries in the world, is there an obligation to fortify basic foods with folic acid, its intake in the normal diet is not usually sufficient. This fact has been proven by many studies”. Note: This thesis mis-cited Murphy and Westmark, 2020 numerous times and cited correctly once. It provided no explanation of the contradictory data.	1,2,3,4,5
Mahalik [39]	All India Institute of Medical Science-Bhubaneswar	India	2024	*BMJ Open*	Prevalence of neural tube defect and its identification during antenatal period: a cross-sectional study in eastern Indian state	“According to some studies, there is 60–100% reduction in the risk of NTD pregnancy after folic acid intervention among pregnant women with prior NTD-affected pregnancies. Observational studies of folic acid efficacy in preventing NTD in pregnant women without a prior NTD-affected pregnancy showed a 0–75% reduction in risk”.	3
Martinez-Morata [40]	Emory University Columbia University	United States	2023	*The Journal of Nutrition, Pre-proof*	Metabolic effects of folic acid supplementation in adults: evidence from the FACT trial	“While adequate folate intake is critically essential for human health and prevention of neural tube defects (NTDs), limited evidence in animal and population studies have associated elevated FA intake with increased risk for cancer, cardiometabolic diseases and neurocognitive endpoints. This suggestive evidence raises questions about potential unanticipated metabolomic effects associated with moderate to high FA intake and UMFA”.	0
Mastrolorito [41]	University of Bari Aldo Moro	Italy	2024	*Chemical Research in Toxicology*	TISBE: A public web platform for the consensus-based explainable prediction of developmental toxicity	“Dev Tox does not have a well-defined biological target being generally capable of affecting all stages of growth from the embryonic to the adolescent age. Keeping this in mind, we first profiled the Dev Tox potential of drugs commonly taken by pregnant women such as folic acid (FA) and progesterone. As shown in the waterfall plots reported in Figures 4 and 5, the calculated SHAP values associated with the chemical structures of these drugs indicate a substantial safety about their Dev Tox risks, and to the best of our knowledge, this is in agreement with clinical evidence and pharmacovigilance studies”. Note: Murphy and Westmark does not show safety of folic acid but does discuss potential risks in vulnerable populations.	4,5
McCaddon [42]	Rutgers University	United States	2023	*Frontiers in Nutrition*	Homocysteine - a retrospective and prospective appraisal	“This fortification program has been highly successful, with reductions in incidence in the United States variously estimated to be 19–40%, depending on the type of NTD and how incidence was counted. Today, more than 80 countries around the world have mandatory folic acid fortification”. Note: There are selective citation and opposite finding errors in stating fortification has been highly successful without discussing contradictory data.	1,4
Melo [43]	Federal University of Sao Carlos	Brazil	2021	*Frontiers in Public Health*	Why are birth defects surveillance programs important?	“The case of folic acid food fortification to prevent neural tube defects demonstrates that. The surveillance data allow for the comparison and monitoring of the prevalence of neural tube defects before, during, and after the implementation of folic acid fortification of staple foods. In different places, these data have been supported the role of folic acid fortification in the decline of neural tube defects birth prevalence, therefore allowing evaluation of the effectiveness of the acid folic fortification as a preventive action”. Note: Murphy and Westmark does not demonstrate a decline in NTDs with folic acid fortification (opposite finding error). Contradictory data are not discussed (selective citation error).	1,4
Menezo [59]	?	France	2022	online	Folic acid (FA) in high doses to improve sperm? A (very) bad idea	“There is no question of doubting the benefit of synthetic folic acid (pteroyl glutamic acid), particularly in avoiding neural tube closure defects. Rather, it is the interest and possible consequences of fortification that are beginning to appear in the scientific literature, particularly with regard to the risks of cancer”.	0
Mohamed [44]	Nursing-Benha University	Egypt	2020	?	Application of health belief model on nursing students’ perception regarding benefits of preconceptional folic acid intake	“Neural tube defects (NTDs) persist as a common and potentially devastating birth defect affecting the central nervous system and axial skeleton that occur as a result of a defect in the closure of a part of the neural tube during the early weeks of intrauterine development”.	3
Mrkulic [45]	University of Sarajevo	Bosnia and Herzegovnia	2021	*European Journal of Biomedical and Pharmaceutical Sciences*	Mothers knowledge and folic acid intake for neural tube defects prevention	“Inadequate folic acid taking can be influenced by various factors, such as: unplanned pregnancy, supplementation costs, maternal age, educational status, marital status, work status, number of previous pregnancies, income, smoker status and level of knowledge, attitude and awareness of the importance of folic acid supplements before and during pregnancy”. “Better socioeconomic status contributes to reducing NTDs prevalence”. “As an important predictor for taking folic acid was mothers’ education similar as in other studies. Although a large number of unemployed mothers in a group of mothers who gave birth to NTD, looking at all mothers’ employment in this study did not affect folic acid taking which are different from other studies”. Note: There is one correct citation regarding better SES reducing NTDs. The serious error is that the major conclusion contradicts Murphy and Westmark, 2020 [1] results, which are not discussed, i.e., the paper states, “In addition it is important to make national recommendations for folic acid supplementation for women of reproductive age, but also change the public health policy that includes mandatory fortification of basic foods as this is the best measure for primary prevention of NTDs in the whole population regardless of employment and education status”.	1,2,4,5
Portilla-Rojas [46]	Pontificia Universidad Javeriana	Columbia	2023	*Birth Defects Research*	A case-control study characterizing polydactyly risk factors in Bogota and Cali, Colombia between 2002 and 2020	“Thus, it is crucial to enhance public health policies aimed at ensuring optimal levels of folic acid prior to embryonic morphogenesis, this can be achieved through direct supplementation or even macroscopic fortification of foods, such as rice. Fortification with folic acid is currently implemented in Colombia, specifically in wheat flour; however, there is no mandatory preconception supplementation”. Note: Murphy and Westmark, 2020 does not talk about preconception supplementation in Columbia (fact not found) and the paper promotes fortification without discussing contradictory findings (selective citation and opposite finding errors).	1,4,5
Pruitt Evans [6]	National Center on Birth Defects and Devel Disabil, CDC	United States	2023	*Epidemiology*	Neighborhood Deprivation and Neural tube Defects	“The United States began fortifying cereal grains with folic acid in 1998 to prevent NTDs in pregnancy.” Note: This paper contains a secondary citation error and failed to cite Murphy and Westmark, 2020 SES data [1], which was directly relevant to their findings, which is a selective citation error.	1,3
Rubin [47]	UCL EGA Institute for Women’s Health	United Kingdom	2023	*PLoS ONE*	Maternal folic acid supplementation and the risk of ankyloglossia (tongue-tie) in infants; a systematic review	“It is difficult to get enough folate from diet alone; around 80 countries, including the USA, Canada, Chile, and South Africa, have mandated the fortification of staple food products, such as corn and wheat, with folic acid to increase folate levels in the population and decrease adverse health outcomes”. Note: The paper also states, “Furthermore, studies analyzing the rates of congenital malformations after mandated wheat fortification suggest that it may be protective against congenital disabilities other than NTDs. In September 2021 the UK government decided to proceed with mandatory fortification of non-wholemeal wheat flour with folic acid. However, this decision has been met with criticism regarding the agreed-upon dosage, which is considered inadequate, …”, without discussing Murphy and Westmark, 2020 contradictory findings.	1,2,4
Shlobin [49]	Northwestern University	United States	2021	*J Neurosurg Pediatr*	Folate fortification and supplementation in prevention of folate-sensitive neural tube defects: a systematic review of policy	“Thirty-two studies evaluated mandatory folate fortification effectiveness only. Twenty-three studies examined spina bifida, 18 anencephaly, 3 encephalocele, and 15 NTDs in general.” Note: Murphy and Westmark, 2020 data are listed in their Table 3 regarding the weak correlation between NTD prevalence & level of fortification regardless of cereal grain fortified and the strong linear relationship between reduced NTDs and higher socioeconomic status; however, there is no discussion of these findings, which contradict their conclusion while papers that agree are discussed.	1,2
Sulistyoningrum [50]	South Australian Health and Medical Research Institute	Australia	2024	*Maternal & Child Nutrition*	Maternal serum unmetabolized folic acid concentration following multivitamin and mineral supplementation with or without folic acid after 12 weeks’ gestation: a randomized controlled trial	“In addition, as NTDs occur in the first month of pregnancy and many pregnancies are unplanned, more than 80 countries, including Australia, Canada and the USA, have mandated fortification of staple foods with FA, further increasing FA intakes of pregnant women”.	0
Taruscio [51]	National Center for Rare Diseases, Italian National Institute of Health (ISS)	Italy	2021	*Medicina*	Multifactorial rare diseases: can uncertainty analysis bring added value to the search for risk factors and etiopathogenesis?	“The overall impact may be moderate on the potential for primary prevention because an adequate folate status is already established as a factor reducing, by approximately 50%, NTD incidence. However, the impact of both epistemic and sampling uncertainties may be severe on the ability to follow up and assess the effects of primary prevention measures. Indeed, a study carried out in the USA recently reported an analysis of a food fortification dataset, highlighting better socio-economic status as a stronger protective factor compared to folic acid fortification”.	0
Tehsin [52]	King Faisal University	Saudi Arabia	2024	*JIIMC*	Awareness of folic acid intake before and during early pregnancy at a primary health polyclinic in Saudi Arabia	“The prevalence of neural tube defects is less than 10 per 10,000 births in countries with folic acid fortification policies”. Note: Murphy and Westmark, 2020 only found less than 10 NTD per 10,000 births in countries with folic acid fortification in the highest SES quintile. The remaining four SES quintiles averaged higher than 10 NTD per 10,000 births”.	4
Thomas [53]	University of Bordeau	France	2023	Dissertation	Nutrition, health and brain structure in aging: epidemiological approach	“Vitamin B9 is provided by green vegetables, legumes and flours supplemented with folic acid in many countries (America including the United States, Australia, etc., since the 1990s) to prevent related neural tube formation defects linked to vitamin B9 deficiency during pregnancy”.	3
Ummah [54]	Istanbul Sabahattin Zaim University	Turkey	2021	*Darussalam Nutrition Journal*	The correlation between folate and vitamin E with anemia incidence of adolescents woman in Islamic boarding school	“Michaela and Cara’s research said that national fortification with folic acid is not associated with a signficant decrease in the prevalence of NTDs at the population level”.	0
Vynckier [55]	KU Leuven, University Hospitals Leuven, St-Augustinus Hospital	Belgium	2021	*Nutrients*	Periconceptional folate supplementation in women after bariatric surgery-a narrative review	“Potential negative effects of high doses of folic acid have been reported. For example, Valera-Gran et al. observed that children whose mothers used folic acid supplements in doses higher than 5000 µg/d during pregnancy had a statistically significant lower mean psychomotor scale score compared to children whose mothers used a more general recommended dose of folic acid supplements (400–1000 µg/day). In a prospective study evaluating the association between maternal multivitamin supplementation, maternal plasma folate and vitamin B12 levels at birth and offspring Autism Spectrum Disorder (ASD) risk in 1257 mother-child pairs, Raghavan et al. reported that low (≤2 times/week) and high (>5 times/week) supplementation was associated with increased risk of ASD. In mothers with very high levels of plasma folate (≥60.3 nmol/L) due to high folic acid intake, resulting in an accumulation of folic acid, as well as very high levels of plasma B12 (≥536.8 pmol/L) at birth, a 2.5 times increased risk of ASD in offspring was observed. This study thus illustrated the concern about extremely high levels of plasma folate and B12 exposure in-utero on early brain development. Wiens and DeSoto advised similar caution regarding the fact that unexpectedly high levels of FA may have implications for proper methylation of DNA during times of rapid cell division, for example, in prenatal development. Whilst the prevalence of neural tube defects has diminished significantly since the start of folic acid fortification in North America in 1998, continued high levels of FA supplementation throughout a pregnancy may not be needed, and are not without risk. In this respect, national folic acid food fortification programs were also challenged by Murphy and Westmark”.	0
Westmark [8]	University of Wisconsin-Madison	United States	2020	*Nutrients*	Reply to “The fallacy of using administrative data in assessing the effectiveness of food fortification initiative dataset. Nutrients 2020, 12, 247”	“The main objective of our paper was to retrospectively address the question of whether folic acid fortification improved NTD at the population level. Toward that goal, we extracted and analyzed relevant data from the Food Fortification Initiative (FFI) dataset. Our study compared countries with national folic acid fortification versus countries without national folic acid fortification, with the rationale that a national fortification policy would directly affect the vast majority of inhabitants of a particular country, despite the importation of grain products from other countries. The primary endpoint of interest examined was the prevalence of NTD. Our overall finding was that “national fortification with folic acid is not associated with a significant decrease in the prevalence of neural tube defects at the population level”.	0
Wojtowicz [56]	Jagiellonian University	Poland	2022	*Ginekologia Polska*	Evaluation of the prevalence of folic acid supplementation before conception and through the first 12 weeks of pregnancy in Polish women at high risk of fetal anomalies	“The best way to check folate status is by measuring maternal red blood cell folate concentrations, because as it was published ≥906 nmol/L (400 ng/mL) is associated with a significantly low risk of NTDs”. Note: we did not categorize this paper in the error group because it did not misrepresent our work although it appears the citation numbering is off. Our paper has no mention of maternal red blood cell folate; however, the paper accurately described our work in the preceding paragraph, “The United States of America (USA) was the first country to implement a national folic acid food fortification program in 1998 to prevent NTDs, and nowadays over 80 countries, but not Poland, fortify food with folic acid. However, questions remain about the impact of food fortification in reducing the occurrence of NTDs. Some authors indicate that there is a very weak correlation between the incidence of NTDs and FA fortification levels, and that a woman’s socioeconomic status may be more influential”.	?
Zhao [58]	Shenyang Medical College	China	2020	*PLoS ONE*	Novel genes associated with folic acid-mediated metabolism in mouse: A bioinformatics study	“This also plays an essential role in the central nervous system, and prevents a neural-tube defect (NTD), which is a major birth defect of the brain and spine that occurs early during the embryonic period”.	3

## Data Availability

The original contributions presented in this study are included in this article; further inquiries can be directed to the corresponding author.

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
