# Peer review of "Bibliometric Analysis and a Call for Increased Rigor in Citing Scientific Literature: Folic Acid Fortification and Neural Tube Defect Risk as an Example"

_nutrients, 2024, doi:10.3390/nu16152503_

Round 1

Reviewer 1 Report

Comments and Suggestions for Authors

I think this is an interesting article that prompts reflection on the misuse of citations and references. Although it is something the entire scientific community is aware of, presenting it in an article is novel. However, this paper lacks coherence and clarity in order to send a clear message. As a reader, I perceived a lack of information (especially in the methodology and results sections), which prevented me from fully understanding what the authors intended to convey.

Abstract:

-   - Folic acid is the synthetic form of folate, thus is not necessary to say “of consuming synthetic folic acid”, in any case, “synthetic form of folate, folic acid”

-     - I think is not necessary to include this sentence: “In 2020, our laboratory published an article entitled, “Folic Acid Fortification and Neural Tube Defect Risk: Analysis of the Food Fortification Initiative Dataset”, in the journal Nutrients”

-          -An objective is needed.

Introduction:

-          -Again “synthetic folic acid” (Line 35)

-          -Line 40: the role of folate? The role of folic acid? Or both?

-          -Lines 42-58: The association between a reduced prevalence of NTD and high SES could be influenced by maternal use of folic acid during pregnancy. People with high SES are more likely to present higher educational level and higher knowledge/worry about them health.

-          -State a clear objective of this work

Methods:

-        -  Very little information. It would be highly advisable to include the sections typical of any review paper and increase the amount of information.

Results:

-     -     Avoid the use of “n=x” in sentences like this: “were found in n=15 publications”

-          -This is not clear. Which “other papers”? – “which is statistically significantly higher than an inaccuracy of 12.3% for all citations from 22 other papers published by the same laboratory in the same timeframe [n=130 non-duplicated full-text publications available: 88% correct; 2.3% mis-citations;10% secondary citation errors]”

-          -In general, results are not clear. Please try to reorganize and better explain your results.

-          -Table 1 is missing

Discussion

-          -The first paragraph is not well linked to the rest of the paper.

-       -   Some interesting points are discussed that are related to your “objective” such as the lack of scientific skills in students. However, in my opinion, all information regarding physiological effects of folic acid are not related to your analyses.

-          -Limitations and strengths are needed.

Thank you

Reviewer 2 Report

Comments and Suggestions for Authors

This is a very interesting and useful manuscript that can inform researchers in nutrition as well as researchers in general, including bibliometricians.

The authors analyzed 50 citations and found that 70% of them were inaccurate, i.e. 35 citations. This is something the scientific community should be aware of. 

The Results can show more data (be more transparent) to better inform the researchers in nutrition field. The authors classified inaccurate citations into 5 categories. My queries for the citation data are as follows:

1. In this manuscript, does one citation means one citing paper? In other words, if a citing paper cited Murphy and Westmark (2020) multiple times, did the authors count it as "one citation"?

2. Were the 5 categories mutually exclusive. In other words, can a citation be simultaneously classified into multiple categories (e.g. secondary citation + incorrect/opposite finding)? If cannot, then the authors should clarify the superiority/priority of the categorization order: e.g. if a citation meets the criteria of category 1, it will be immediately coded as 1; if not, see if it meets the criteria of category 2 and coded as 2; if not... etc.

3. Please show the numbers (N) for each inaccurate citation category. Contextually, what are the most common inaccurate citations? E.g. wrongly cited some % or numbers, distorted the conclusion of Murphy and Westmark (2020), or quoted a minor finding of X but ignored a major result of Y? These pieces of information will enrich the Results.

4. The citation statements of the citing papers should be provided in a supplementary file (e.g. Excel sheet), so that readers can grasp the essence of the inaccuracies. Researchers can cite more correctly in the future after seeing the data.

5. "It is also disturbing that 7 of the citations were from trainee theses or dissertations, which all contained inaccuracies (5 serious errors and 2 secondary citation errors)." This is a good finding. It should be reported in the Results first, then elaborated in the Discussion. Meanwhile, it means that trainee documents had a 70% serious error rate whereas the overall dataset had a 50% serious error rate. Were these numbers significantly different? Please test it and report it in the Results. If statistically significantly different, it is more convincing to say it is "disturbing".

The rest of the Discussion part is very informative to the readers. Congratulations to the great work.

Round 2

Reviewer 1 Report

Comments and Suggestions for Authors

In general, I am not clear that this work is a bibliometric analysis.

Methods: Line 95: I think you mean unavailable.

Methods: It is still very sparse. You need to include all the sections of a bibliometric study (study design, search strategy…).

Methods: Where and for what did you use Fisher?

Author Response

Thank you for the re-review of our manuscript. Edits are highlighted in yellow in the WORD doc.

In general, I am not clear that this work is a bibliometric analysis. We have included a copy of the BIBLIO checklist as Table S1, which contains line numbers referencing the inclusion of elements for a bibliometric analysis.

Methods: Line 95: I think you mean unavailable. We meant "available". We have added text to indicate the "duplicate" publications (preprint plus publication were available).

Methods: It is still very sparse. You need to include all the sections of a bibliometric study (study design, search strategy…). We have revised the section titles and inserted, "The study design was a content analysis...". We also added the phrase, "The search strategy involved..." before the description of the search engine, etc. We're confused on what additional specific information you would like included so that an independent party could replicate the study.

Methods: Where and for what did you use Fisher? Fisher exact tests were used for all of the reported p values in the paper.

Reviewer 2 Report

Comments and Suggestions for Authors

The authors have satisfactorily addressed my concerns.

Author Response

The authors have satisfactorily addressed my concerns.

Thank you.